# Acute Physiological Response to Different Sprint Training Protocols in Normobaric Hypoxia

**DOI:** 10.3390/ijerph19052607

**Published:** 2022-02-24

**Authors:** Naomi Maldonado-Rodriguez, David J. Bentley, Heather M. Logan-Sprenger

**Affiliations:** 1Faculty of Kinesiology & Physical Education, University of Toronto, Toronto, ON M5S 1A4, Canada; nmrod@mail.ubc.ca (N.M.-R.); bentley.dj@gmail.com (D.J.B.); 2Canadian Sport Institute of Ontario, Toronto, ON M1C 0C7, Canada; 3Faculty of Health Sciences, Ontario Tech University, Oshawa, ON L1H 7K4, Canada

**Keywords:** hypoxia, sprint training, physiological response

## Abstract

Background: the purpose of this study was to examine acute physiological responses to and the performance effects of two sprint training protocols in normobaric hypoxic conditions. Methods: Healthy competitive female (*n* = 2) and male (*n* = 5) kayakers (19 ± 2.1 years) performed four sprint training sessions on a kayak ergometer over a period of two weeks. Participants performed five sets of 12 × 5 s sprints or 3 × 20 s sprints in both normobaric normoxic (NOR, F_i_O_2_ = 20.9%) or normobaric hypoxic (HYP, F_i_O_2_ = 13.6%) conditions. The peak power output (PPO), rate of perceived exertion (RPE), and heart rate (HR) of each participant were monitored continuously. Their blood lactate concentrations ([BLa^+^]), in addition to their blood gas (mixed-venous partial pressure (p) of carbon dioxide (pCO_2_), O_2_ (pO_2_), and oxygen saturations (sO_2_)) were collected before and after exercise. Results: A significantly greater RPE, HR, and [BLa^+^] response and a significant decrease in pCO_2_, pO_2_, and sO_2_ were observed in HYP conditions versus NOR ones, independent of the type of training session. The PPO of participants did not differ between sessions. Their RPE in HYP12 × 5 was greater compared to all other sessions. Conclusions: The HYP conditions elicited significantly greater physiological strain compared to NOR conditions and this was similar in both training sessions. Our results suggest that either sprint training protocol in HYP conditions may induce more positive training adaptations compared to sprint training in NOR conditions.

## 1. Introduction

Traditional “altitude training”, where athletes live and train at real or simulated altitudes of 1800–2400 m, has focused on enhancing sea level endurance performance by improving red cell mass [1]. Live high train low (LHTL) models are commonly used in preseason training to enhance aerobic and potentially anaerobic performance [2,3]. LHTL training approaches require athletes to adhere to prolonged and consistent exposure (≥3 weeks) to hypoxia, while training at sea level, in order to stimulate an increase in red blood cell (RBC) mass whilst reducing the potentially detrimental effects of continuous altitude exposure on detraining [4]. However, such training is neither time- nor cost-effective [5]. While the positive effects on aerobic capacity are well-documented, the reported effects on anaerobic performance have been equivocal.

Traditional LHTL protocols, designed to induce improvements in aerobic capacity, typically do not result in meaningful anaerobic improvements, most likely as a result of training design that does not sufficiently stress the anaerobic system [6,7]. Recently, intermittent hypoxia training (IHT), “a method where athletes live at or near sea level but train under hypoxic conditions”, has been explored as a potential alternative to traditional training at altitude to stimulate adaptations that may affect anaerobic performance [7]. Studies have shown that sport-specific IHT may lead to speed/power performance improvements or repeated sprint ability via increased resting pH, enhanced buffering capacity, and improved blood perfusion [7,8,9,10,11,12,13,14,15]. However, despite many potential physiological adaptations, IHT has failed to show consistent performance improvements, such as in relation to power output or performance in 10 s or 30 s all out tests [13,16,17]. One important consideration is that training must be performed at a maximal intensity in order to augment anaerobic performance [5,7,15]. One way of adapting IHT for anaerobic performance-specific adaptations may be sprint training in hypoxia [7]. The parameters of training design (e.g., modality, work-to-rest ratio, repetition duration) should be such that they result in incomplete recovery, so that they induce glycolytic adaptations. One such parameter, sprint duration, largely influences the relative contribution of aerobic and anaerobic energy systems. A duration of 30 s is conventionally used in sprint training, with the first 5–10 s representing the period during which peak power is achieved, with the latter 20–25 s constituting an effort to maintain this output [18]. Hazell and colleagues (2010), when comparing 10 s and 30 s sprint protocols, suggested that the initial generation of peak power output (PPO) was likely responsible for sprint interval training adaptations [19]. Sprint training is primarily dependent on the ability to match ATP resynthesis rate to utilization rate and altered intracellular and extracellular ion concentrations [12]. The added stimulus of hypoxia is likely to increase metabolic stress and, thus, hypothetically induce greater physiological adaptations and performance improvements.

The sport of flat water canoeing and kayaking encompasses events ranging between a ‘sprint’ (200 m) of <30 s to events that are 1–2 min in duration (1000 m). It has been shown that there is a large aerobic component to these events, especially 1000 m events [20]. In contrast, in the sprint event, there is a heavy anaerobic component, with athletes requiring considerable upper body anaerobic capacity. Competitive canoe and kayak athletes undertake sprint training and other forms of anaerobic conditioning in their preparation [20].

The acute response to different durations of anaerobic work intervals, with respect to work volume and work-to-rest ratio, performed in hypoxia, has not been previously investigated, and this is especially true in the context of kayak athletes, for whom anaerobic training is an important sport-specific consideration. The purpose of this study was to examine acute physiological responses to two sprint training protocols (sets of 5 s versus 30 s) in normobaric hypoxic conditions. We hypothesized that, (1) the hypoxic condition would induce greater physiological strain compared to normoxic conditions, and (2) longer duration sprints would result in a greater physiological response in terms of blood parameters.

## 2. Materials and Methods

Seven healthy well-trained male or female kayakers (female: *n* = 2; male: *n* = 6) aged 15–24 (mean ± SD: 19 ± 2.1 years) were recruited from local canoe–kayak clubs (training 18 ± 2 h per week). All kayakers competed at a national and international level as part of their respective provincial or national federations in flatwater kayaking. Participants were sea level natives with no exposure to hypoxia training. All participants were actively training athletes in their off-season. Prior to the initial training sessions, a risk assessment was completed to identify any potential confounding respiratory, neurological, musculoskeletal, or circulatory conditions that may have been deemed a risk to sprint training or training in hypoxia. Parents (assent was required) and participants were informed both verbally and in writing of the experimental protocol and potential risks before giving their verbal and written assent and consent, respectively, to participate. The Research Ethics Board at the Canadian Sport Institute Ontario approved the study (REB#18-01). 

### 2.1. Experimental Procedure 

Athletes completed four sprint training sessions on a kayak ergometer (K1 Speed Stroke Ergometer, KayakPro, Miami Beach, FL, USA) over a period of 2 weeks, with a minimum of 48 h between each session. Each session occurred in an environmentally controlled chamber (K2 Room, Storex Ca Inc., Montreal, QC, Canada). In a single-blind fashion, each session was performed in the altitude chamber to ensure participants were blind to the environmental condition. The order of the training sessions was randomized to control for potential physiological adaptations that may have occurred throughout the duration of the testing. Previous studies have demonstrated that participants are not able to distinguish between hypoxia and normoxia during intermittent sprint training [7]. The trials were conducted at the same of time of the day. Any supplementary normoxic training that occurred outside the study was recorded (modality, duration, and intensity). 

### 2.2. Training Protocol 

Participants performed five sets of 12 × 5 s sprints (protocol 1) or 3 × 20 s sprints (protocol 2) in both normobaric normoxia (NOR, 0 m altitude, F_i_O_2_ = 20.9%) and normobaric hypoxia (HYP, 3500 m altitude, F_i_O_2_ = 13.6%) (see Appendix A). The warm-up consisted of 5-min of light intensity paddling at a self-selected pace, followed by 4 × 10 s preparatory ‘submaximal efforts’ with 20 s of light intensity work between each bout and 2 min of passive rest before the start of the sprint session (11 min total). Protocol 1 consisted of five sets of 12 × 5 s sprints with 15 s of active recovery between each repetition and 2 min of passive rest between sets. Protocol 2 consisted of five sets of 3 × 20 s sprints with 60 s active recovery between repetition and 2 min of passive rest between sets. Both protocols were matched for total work volume and work-to-rest ratio (1:3) based on recommendations set forth by Brocherie and colleagues (2017) [8]. The focus of the training was to maximize hypoxia exposure in order to improve anaerobic capacity by way of short high-intensity exercise, replicating the demands of kayaking. Athletes were asked to perform maximally for each sprint. 

Training was completed in a normobaric hypoxic chamber that was purpose-built for intermittent hypoxia training (K2 Room, Storex Ca Inc., Montreal, QC, Canada). The chamber is a 5 × 5 m room with an airlock and a glass wall for viewing access. The temperature and humidity were set at approximately 20 °C and 20%, respectively, for consistency. Participants were monitored continuously using heart rate (HR) and rate of perceived exertion (RPE) [21]. Their peak HR was captured after every sprint using a Polar Strap (Polar H10, Polar Electro, Nassau, NY, USA) and recorded using the FIT IV Pulse application on an iPad (iPad version 3, Apple, Cupertino, CA, USA). Their RPE was recorded after every set on a scale of 6 to 20. Capillary blood samples for [BLa^+^], blood gas, and metabolite analysis were collected throughout. [BLa^+^] was collected pre-exercise, after set 3, and post-exercise. Blood gas and metabolites were collected immediately pre- and post-exercise. 

### 2.3. Blood Sampling

Blood samples (5 uL) were collected from participants’ fingertips and analyzed for blood lactate concentration [BLa^+^] using a portable hand-held blood lactate analyzer (Lactate PRO, USA) pre-exercise (5 min prior to start), after set 3, and post-exercise (5 min post). Blood bicarbonate (HCO3^−^), pH, mixed-venous partial pressure of carbon dioxide (pCO_2_), oxygen (pO_2_), and oxygen saturation (sO_2_) were measured via capillary analysis using a blood gas analyzer (ABL80, Radiometer, Mississauga, ON, Canada). Various metabolites, including sodium (Na^+^), potassium (K^+^), calcium (Ca^2+^), and chloride (Cl^−^), were also assessed using a 125 uL plastic capillary tube with 70iu balanced heparin pre- and post-exercise and measured using a blood gas analyzer (ABL80, Radiometer, Mississauga, ON, Canada). All resting blood samples were taken in normoxia while the post-exercise blood draw was collected in either HYP or NOR conditions depending on the exercise session. 

### 2.4. Performance Measures 

The participant’s peak power output (PPO) for each sprint was recorded using the kayak ergometer’s integrated monitor. The PPO was then averaged per set and per session and normalized to body mass (W/kg). 

### 2.5. Statistical Analysis 

Data are mean ± standard deviation (SD), unless otherwise stated. Changes in the mean and standard deviation of the variables representing between- and within-subject variability were assessed using a two-way repeated-measures ANOVA in SPSS (Version 24, IBM Corp., Armonk, NY, USA). RPE and peak HR were averaged to a session mean for all four trials. The difference between pre- and post-exercise values for [BLa+], blood gas, and metabolites was calculated to obtain a mean change over time, thus removing the third factor of time. Only mean values in PPO were compared using a three-way ANOVA with repeated measures model. Post-hoc analyses were conducted where applicable and adjusted for multiple comparisons using Bonferroni correction. Both HR and sO_2_ were not normally distributed; as both did not respond to transformation, data analysis was run with and without outliers (one in each). As they did not change, results from analysis with outliers are reported. To supplement important findings, effect sizes (η^2^) were calculated as the ratio of the mean difference to the pooled SD of the difference. The magnitude of the effect size was classed as trivial (<0.2), small (0.2–0.6), moderate (0.6–1.2), large (1.2–2.0), and very large (>2.0) based on previous published guidelines [22]. Moreover, exact *p* values and Cohen *d* are presented to show the magnitude of effect. 

## 3. Results

The mean ± SD values are outlined in Table 1. The PPO of participants did not differ between training protocols (Table 1). All participants trained at a very high intensity, with mean session RPEs ranging from 15 to 17 (Table 1). A significant interaction was found between the environmental condition and training protocol with respect to RPE (*p* = 0.011, η^2^ = 0.76). Post-hoc analyses revealed that participants reported a significantly greater RPE after the HYP12 × 5 protocol, compared to the NOR12 × 5 protocol (*p* = 0.003). In addition, we found a significant interaction in terms of peak HR between training protocols (12 × 5 vs. 3 × 20) and environmental conditions (normoxia vs. hypoxia) (Table 1, *p* = 0.014, η^2^ = 0.73). Peak HR in both HYP12 × 5 and HYP3 × 20 was greater than its normoxia counterpart, with *p* = 0.029 and *p* = 0.025 respectively. Peak HR in NOR3 × 20 was significantly greater than in NOR12 × 5 (*p* = 0.003). There were no differences in terms of hypoxia between training protocols. 

Post-exercise [BLa^+^] values were elevated in all training protocols, compared to pre-exercise resting values (refer to Table 1). Mean [BLa^+^] values were above the lactate threshold, which is generally defined as between 4 mmol/L [20]. Post-exercise [BLa^+^] levels were significantly greater in hypoxia, compared to normoxia (*p* = 0.029, η^2^ = 0.65), regardless of the training session (12 × 5 or 3 × 20) (Table 1). Additionally, post-exercise [BLa^+^] levels were significantly greater following the 3 × 20 protocol, compared to the 12 × 5 protocol, regardless of the environmental condition (*p* = 0.016, η^2^ = 0.72).

No significant differences were observed in pH between conditions or sprint sessions (*p* > 0.05). However, we did note an interaction that trended towards significance (*p* = 0.085). For both the 3 × 20 sprint sessions (HYP and NOR), pH fell below 7.35, suggesting that the 3 × 20 sprint session elicited metabolic acidosis in both HYP and NOR environmental conditions. Post-exercise pCO_2_ was significantly greater than pre-exercise values in the hypoxia conditions only (*p* = 0.05, η^2^ = 0.57), independent of the training session (Figure 1A,B). On the other hand, post-exercise pO_2_ was significantly reduced in the hypoxia session (*p* = 0.001, η^2^ = 0.91), irrespective of the training session (Figure 1C,D). There was no interaction between training protocol and environmental condition in either blood gases. Additionally, sO_2_ was significantly lower in hypoxia than in normoxia, independent of the training protocol (*p* = 0.001, η^2^ = 0.9, Figure 1E,F). The training protocol (12 × 5 or 3 × 20) had no statistical effect on pCO_2_ or pO_2_ response. Lastly, we found no significant changes in the blood metabolites measured (Na^+^, K^+^, Ca^2+^, Cl^−^, HCO_3_^−^). While it did not reach significance, HCO_3_^−^ showed a trending decrease (*p* = 0.067, η^2^ = 0.52) in the NOR3 × 20 and both hypoxia sessions. 

## 4. Discussion

The results of this study demonstrate that regardless of the sprint training stimulus (12 × 5 vs. 3 × 20), a controlled normobaric hypoxic environment of 13.6% O_2_ elicited significantly greater acute responses in terms of RPE, HR, BLa concentration, and pCO_2_, with significantly lower post-exercise pO_2_ and sO_2_, along with a trend of lower blood pH and HCO3^−^, compared to a sprint training session in normobaric normoxic conditions. When comparing the sprint training session stimulus (12 × 5 vs. 3 × 20), the results show that the RPE of the participants was higher in the 12 × 5 sprint workout; however, their HR and BLa^+^ concentrations were significantly greater in the 3 × 20 sprint workout, with no difference between the sprint training stimulus in terms of blood gases, pH, or blood metabolites. 

### 4.1. Peak Power Output during Sprint Training in Hypoxia and Normoxia

Athletes were able to maintain their PPO, independent of the training session. This is congruent with the literature, which suggests that PPO is generated during the first 5–10 s of exertion [19,23]. Thus, athletes would have had time to reach their maximal power output even during the short-duration sprint protocol. An athlete’s ability to generate power throughout a session is an important consideration when it comes to potential adaptations to the anaerobic system. It is well-documented that athletes must train at maximal intensities in order to induce improvement in anaerobic capacity, likely due to enhanced glycolytic activity in muscles via increased mRNA expression related to pH regulation, upregulation of anaerobic metabolism, increased buffering capacity, modified fast twitch (FT) fiber behavior, and increased end-product metabolite removal mechanisms [5,7,8,13,14,15,24,25]. In contrast, a reduction in PPO may have indicated that the training stimulus was too high. This has historically been an issue in this field of research, where prolonged exposure to hypoxia leads to a reduction in training quality and intensity, which may explain the lack of performance improvements previously noted, despite physiological adaptations [26,27,28]. It is thus paramount that athletes work maximally. Our results demonstrate that the sprint prescription (12 × 5 or 3 × 20) did not impair repeated maximal power output over a single session. The question becomes whether the repeated maximal power output can be sustained over multiple sessions to elicit positive training adaptations. 

### 4.2. Physiological Responses to Sprint Training in Hypoxia and Normoxia

Peak HR was greater in hypoxia than normoxia, independent of the training session. As expected, the reduction in the fraction of inspired oxygen (F_i_O_2_ = 13.6%) resulted in greater cardiovascular strain, which translated into an increase in HR. Moreover, peak HR in NOR3 × 20 was greater than in NOR12 × 5. Additionally, athletes rated the intensity of the exercise as very high. Only HYP12 × 5 resulted in a greater RPE compared to its normoxia counterpart, but this was not reflected in any PPO differences. Interestingly, no differences were noted between NOR3 × 20 and HYP3 × 20. Studies have reported that shorter sprint durations see a greater relative contribution from the anaerobic system [29]. Thus, a 5 s sprint would be a greater stressor on the anaerobic system compared to a 20 s sprint. This trend is also seen in [BLa^+^], which was greater in HYP12 × 5 than in NOR12 × 5. Post-exercise [BLa+] values were greater than pre-exercise values, regardless of the training session. These results are not surprising, as the athletes were instructed to sprint maximally. A high lactate accumulation is consistent with the literature, which has shown that high intensity short duration exercise, such as sprint training, leads to [BLa^+^] accumulation above the lactate threshold (typically defined as 4 mmol/L). Elevated lactate levels at the cessation of exercise is indicative of anaerobic metabolism [30]. Additionally, [BLa^+^] was higher in HYP12 × 5 compared to NOR12 × 5, but no differences were noted between NOR3 × 20 and HYP3 × 20. This appears to be the only physiological response that supports our RPE findings. Much of our data suggests that the acute physiological response to HYP12 × 5 and HYP3 × 20 was similar. It is possible that the increase in [BLa+] resulted in greater muscular discomfort and, consequently, an increase in RPE and [BLa+] [31,32].

### 4.3. Blood Gas Responses to Sprint Training in Hypoxia and Normoxia

A significant decrease in pCO_2_ and pO_2_ was observed in the hypoxic conditions. However, this response did not differ between the training protocols. A decrease in pCO_2_ is in accordance with a left shift of the oxyhemoglobin curve, which is typically observed following acute exposure to hypoxia [33]. The increase in ventilation, and, resulting respiratory alkalosis, are the body’s attempt to increase O_2_ saturation in an environment with reduced oxygen availability. This reduction in pCO_2_ may also reflect increased buffering activity, whereby, in an attempt to maintain a stable pH, the body will employ various mechanisms to buffer and maintain its acid-base balance. However, given the fact that HCO_3_^−^ did not decrease significantly, ventilation may have played a greater role in the pCO_2_ response. Nonetheless, it is more likely that both contributed to this response. On the other hand, a decrease in mixed-venous pO_2_ is a typical response to high-intensity exercise, as oxygen consumption increases at the tissue level. The magnitude of this response is likely a reflection of hypoxic conditions. It is also important to note that in normoxia, the mean post-exercise O_2_ values increased (non-significantly) compared to pre-exercise. Elevated pO_2_ levels following exercise are indicative of hyperventilation and are congruent with our pCO_2_ results [25]. The reason why pO_2_ decreased in hypoxia is unclear and may be related to the magnitude of the hypoxic stimulus delivered [34]. Similarly, sO_2_ significantly decreased in hypoxia compared to normoxia, independent of the training session. This supports the literature, which has established that sO_2_ decreases acutely in hypoxia, reflecting the lower F_i_O_2_ and the increase in oxygen extraction during exercise [14,25]. 

While pH did not reach statistical significance, the mean post-exercise values suggest that metabolic acidosis (pH < 7.35) was present or very close in three of the training protocols (HYP12 × 5, NOR3 × 20, and HYP3 × 20) but not in NOR12 × 5 (pH = 7.39) [35]. This is possibly a result of the body’s inability to eliminate or buffer waste metabolites or [BLa+], thereby leading to the accumulation of the measured metabolites and a decrease in pH [36]. Given the fact that one important adaptation induced from training in hypoxia is an increase in resting pH, it is possible that this acute response may indicate that this protocol provided an adequate stressor to stimulate changes. These results clearly demonstrate that sprint training in hypoxia resulted in greater physiological strain compared to normoxia, with no differences seen between the training protocols. 

### 4.4. Perceptual Response to Sprint Training in Hypoxia and Normoxia

It is well-documented that many anaerobic performance improvements, such as speed or power development, are mediated via adaptations in the neuromuscular system by way of motor unit recruitment, activation, and firing [37,38,39,40]. Studies show that maximal effort sprints require high levels of motor unit activation [40,41]. In this study, participants perceived the HYP12 × 5 session to be harder than the three other sessions, despite them resulting in a similar PPO. One potential reason for this may be that the greater repetition of sprints in this session produced a greater stressor on the neuromuscular system due to rapid acceleration and deceleration. There is evidence that neural fatigue may be caused by a decrease in reflex sensitivity, which has been associated with force production and propulsion [18,42]. Bowtell and colleagues (2014) noted significantly reduced iEMG activity and running speed following sprint training in hypoxia [14]. These changes in neuromuscular activity may reflect fatigue development, possibly due to reduced central neural drive or impaired neuromuscular transmission [14,43]. Moreover, metabolic changes in the muscle, such as the [BLa+] accumulation and the decrease in pH seen in this study, may have accelerated the onset of muscle fatigue [40]. It is worth noting that in severely hypoxic conditions, neural fatigue may be a limiting factor and prevent athletes from training at the maximal intensities needed to obtain performance increments [3,44]. As such, the interaction between training protocol and altitude should be carefully considered when designing a program to achieve a stimulus great enough to induce adaptations but that does not impair performance. 

### 4.5. Limitations 

This study included a relatively small sample size, which may partly explain why certain effects were observed and others were not. Additionally, the results may have been affected by the range of competition levels tested. Participants ranged from national level to internationally ranked athletes. However, research shows that elite and sub-elite athletes often have different strengths and anaerobic power profiles [45,46]. Therefore, testing a sample of athletes of mixed competition levels may have confounded the results, especially given the small sample size. Moreover, although an effort was made to standardize the encouragement given, it is possible that the encouragement provided varied between athletes and may have affected the PPO results [47]. We also did not control the athlete’s training outside of the four sessions. We asked athletes to record the modality, intensity, and duration of their training and maintain their training schedules consistently throughout the study. However, adherence to instructions is not always high and it is possible that supplementary training affected the athlete’s ability to perform maximally.

### 4.6. Future Directions and Practical Application

Ultimately, our results suggest that sprint training in hypoxia elicits a greater physiological response compared to training in normoxic conditions, irrespective of the training protocol. However, athletes found HYP12 × 5 more difficult, despite recording similar physiological responses to other sessions. Understanding the reasons why athletes’ perceived rate of exertion was higher in this training protocol, while eliciting a similar physiological response, has important implications for training prescription. Special care should be taken when selecting a protocol that is applicable to the sport in question (i.e., which considers the demands of the sport) and one that considers the contribution of metabolic systems. Lastly, given the cross-sectional design of this study, we are only able to comment on acute physiological responses. Future research should explore the physiological response and adaptation to different training protocols in hypoxia using a longitudinal study design.

## 5. Conclusions

The results of this study indicate that many of the acute responses seen were associated with the environmental conditions. Overall, the hypoxic conditions were physiologically more stressful than the normoxic conditions, irrespective of the training protocol, which is in keeping with our hypothesis. The significant changes observed in RPE, [BLa+], pCO_2_, pO_2_, and sO_2_, as well as the non-significant but potentially meaningful changes seen in pH and HCO_3_^−^, indicate that the hypoxia sessions were more physiologically stressful for the athletes. It does not appear that the type of training influenced performance or physiological response, except in the case of [BLa+]. 

## Figures and Tables

**Figure 1 ijerph-19-02607-f001:**
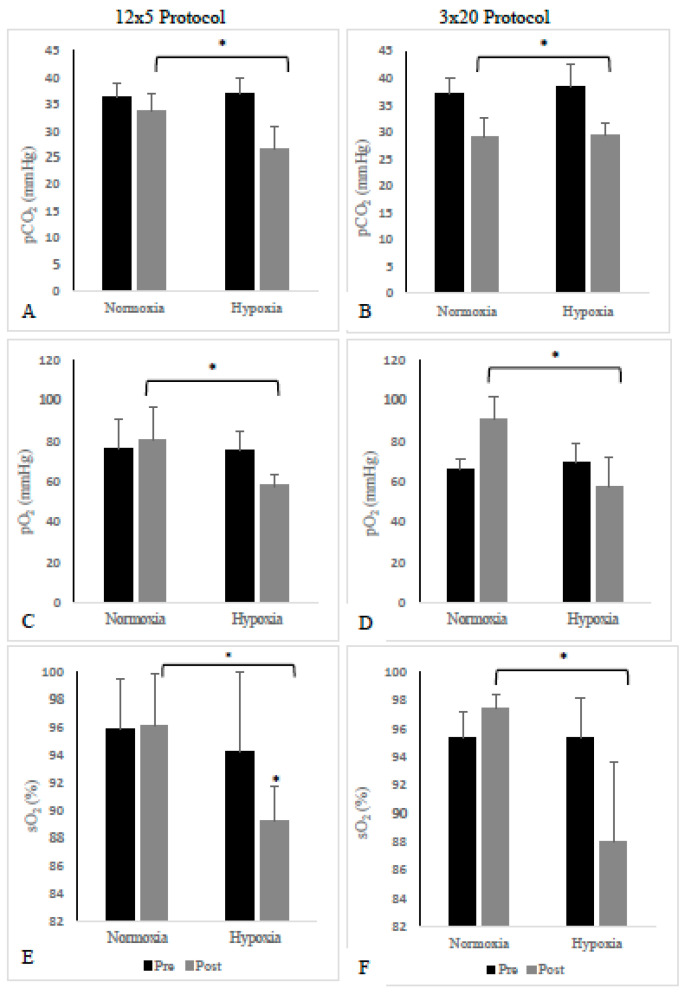
Mean pre- and post-exercise blood gas measurements for the two sprint training protocols (12 × 5 versus 3 × 20). (**A**) partial pressure of carbon dioxide (pCO_2_) in the 12 × 5 protocol; (**B**) partial pressure of carbon dioxide (pCO_2_) in the 3 × 20 protocol; (**C**) partial pressure of oxygen (pO_2_) in the 12 × 5 protocol; (**D**) partial pressure of oxygen (pO_2_) in the 3 × 20 protocol; (**E**) blood oxygen saturation (sO_2_) in the 12 × 5 protocol; (**F**) blood oxygen saturation (sO_2_) in the 3 × 20 protocol. Data are mean ± SD. * Significant difference versus normoxia, independent of training protocol (*p* < 0.05).

**Table 1 ijerph-19-02607-t001:** Mean performance and physiological measures pre- and post-training.

		12 × 5 Protocol		3 × 20 Protocol	
		NOR	HYP	NOR	HYP
PPO (W/kg)		3.87 ± 1.04	3.84 ± 0.89	3.56 ± 0.32	4.14 ± 1.4
RPE		15 ± 1.21	17 ± 0.89 *	16 ± 0.73	16 ± 1.35
Peak HR (bpm)		159 ± 11 ^	167 ± 10 *	164 ± 8	170 ± 11 *
[BLa^+^] (mmol/L)	Pre	1.18 ± 0.2	1.9 ± 0.57	1.64 ± 0.6	1.44 ± 0.48
	Post	4.23 ± 1.74	8.52 ± 2.5 *	9.02 ± 3.8 ^	10.28 ± 3.0 *
Blood Gas/Metabolites					
pH	Pre	7.41 ± 0.042	7.41 ± 0.013	7.43 ± 0.031	7.42 ± 0.035
	Post	7.39 ± 0.051	7.36 ± 0.037	7.31 ± 0.010	7.34 ± 0.072
pCO_2_ (mmHg)	Pre	36.33 ± 2.66	37.00 ± 2.92	37.00 ± 3.10	38.33 ± 4.32
	Post	33.67 ± 3.33	25.8 ± 4.44 *	29.00 ± 3.39	29.33 ± 2.34 *
pO_2_ (mmHg)	Pre	76.33 ± 14.26	75.4 ± 9.26	65.8 ± 5.17	69.17 ± 9.79
	Post	80.33 ± 16.75	57.8 ± 5.45 *	90.80 ± 11.26	57.17 ± 14.74 *
HCO_3_^−^ (mEq/L)	Pre	22.6 ± 2.90	23.20 ± 1.63	24.38 ± 1.95	24.25 ± 1.71
	Post	19.98 ± 2.69	14.18 ± 2.30	14.6 ± 4.81	15.7 ± 3.14
sO_2_ (%)	Pre	95.8 ± 3.6	94.25 ± 5.75	95.28 ± 1.9	95.28 ± 2.84
	Post	96.07 ± 3.7	89.2 ± 2.52 *	97.43 ± 1.0	87.97 ± 5.6 *

Data are mean ± SD. NOR, normobaric normoxia (20.9% O_2_); HYP, normobaric hypoxia (13.6% O_2_); PPO, peak power output; W/kg, watts per kilogram body mass; HR, heart rate; bpm, beats per minute; [BLa^+^], blood lactate concentration; pCO_2_, partial pressure of carbon dioxide; pO_2_, partial pressure of oxygen; HCO_3_^−^, bicarbonate; sO_2_, saturation of oxygen. * Significant difference between the normoxic and hypoxic response within the training protocol (*p* < 0.05); ^ Significant difference between training protocols (*p* < 0.05).

## Data Availability

The data presented in this study are available on request from the corresponding author. The data are not publicly available due to privacy restrictions.

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
