# Peer review of "Acute Physiological Response to Different Sprint Training Protocols in Normobaric Hypoxia"

_ijerph, 2022, doi:10.3390/ijerph19052607_

Round 1

Reviewer 1 Report

The authors need to be commended on the well written paper.  They are clearly knowledgeable on the topic and have designed and executed a nice study which furthers the understanding of IHT.  I have only some minor amendments to make in the methods and results, but otherwise it was really impressive to read. 

Minor amendments:

Training Protocol section on line 4, need to put 4 x 10s

20° needs to be 20°C

I would change the terminology of subjects to participants throughout

You need to give the model of the polar heart rate monitor and the location details etc.

A reference is needed for the Borg Scale.

You mention towards the end of the methods that sleep quality and duration was measured, but how? there is no mention of this in the results. 

How was motivation measured?

Can you be more specific in terms of time when you say samples were taken pre and post exercise. Would this be 5 minutes? 2 minutes? 30 minute?

With regards to the results, I would take the gridlines off Figure 1 and label the graphs A, B, C etc.  

Author Response

Thank-you for your comments.

Reviewer 1:

The authors need to be commended on the well written paper.  They are clearly knowledgeable on the topic and have designed and executed a nice study which furthers the understanding of IHT.  I have only some minor amendments to make in the methods and results, but otherwise it was really impressive to read. 

Minor amendments:

Training Protocol section on line 4, need to put 4 x 10s

Thank-you! The change has been made.

20° needs to be 20°C

Thank-you! The change has been made.

I would change the terminology of subjects to participants throughout

Thank-you! The change has been made.

You need to give the model of the polar heart rate monitor and the location details etc.

Thank-you for pointing this out. The change has been made.

A reference is needed for the Borg Scale.

Thank-you. The reference below has been included:

Borg GA. Psychophysical bases of perceived exertion. Med Sci Sports Exerc. 1982;14(5):377–381. PubMed ID: 7154893 doi:10.1249/00005768-198205000-00012

You mention towards the end of the methods that sleep quality and duration was measured, but how? there is no mention of this in the results. 

Sleep duration and quality were measured through a daily questionnaire.  We have decided to remove this sentence from the methods to avoid confusion as we decided not to include the findings within the Results section.

How was motivation measured?

Motivation was measured through a subjective questionnaire of perceived readiness and motivation to train prior to each training session. We have decided to remove this sentence from the Methods to avoid confusion since the results of the questionnaire are not included in the Results section of this paper.

Can you be more specific in terms of time when you say samples were taken pre and post exercise. Would this be 5 minutes? 2 minutes? 30 minute?

Thank-you for your comment. All pre-exercise samples were taken 5 minutes before the training session.  Post-exercise samples were taken 5 minutes after the completion of the training session. New language has been added to the Methods to make this more clear;

Blood samples (5 ul) were collected from fingertip and analyzed for blood lactate concentration [BLa+] using a portable hand-held blood lactate analyzer (Lactate PRO, Arkray, Japan) at pre-exercise (5 minutes prior to start), after set 3, and post-exercise (5 minutes post).”

With regards to the results, I would take the gridlines off Figure 1 and label the graphs A, B, C etc.  

Thank-you for your comment. We have taken the gridlines off Figure 1 and labelled appropriately as suggested.

Reviewer 2 Report

Thank you for the opportunity to review this publication. The submitted study "aims to examine the acute physiological responses 7 and performance effects of two sprint training protocols in normobaric hypoxic conditions" 

The question of the study is definitely interesting for a target group of athletes, caregivers, sports scientists and medical professionals.

The manuscript provides an interesting insight on the topic. However, concerns on some key points are listed below:

INTRODUCTION

The authors present a complete introduction on the physiological adaptations of training in hypoxic conditions. However, it is necessary to make a detailed explanation of the sport practiced by seven kayakers. Knowing in detail the physiological and competitive requirements of this sport can help understand the adaptations during the study. According to the literature, it seems that not all adaptations in all types of athletes are the same.

The paper does not clearly explain its advantages concerning the literature: it is not clear the novelty and contributions of the proposed work: does it propose a new method? Or does the novelty only consist of the application?

MATERIALS AND METHODS

Eight participants are a very low number of participants. Please add information regarding the sample calculation and the power of your analysis.

There is no statement of parental consent concerning the underage kayakers participated in the study?

What is the study design? Please use a simple diagram or figure to illustrate this paper's whole idea, and the modification it has been made from previous work or traditional framework.

In addition, it is necessary to defend the type of design according to the number of subjects and other variables to verify the validity of the study.

What guidelines should athletes follow during protocol sessions and training? Were they only informed both verbally and in writing of the experimental protocol and potential risks? Were measurements made at the same time of day? What about in relation to training? Were there instructions to not train the day of testing prior to assessment? What about hydration status, was this measured prior to assessments?

Why did you only analyze peak power output and not mean power output or total distance?

L138: L94: SPSS v.24 manufacturer details: IBM Corp., Armonk, NY.

"Cohen d are presented to show the magnitude of effect". The authors must specify the thresholds, reference them in "Statistical Analysis" and detail them in all the results.

RESULTS

Please edit the table according to the journal format.

All the data in text and table must go with a decimal, except the values of "p".

The font of the figures must match the font of the text. Also include the vertical axis line and consider removing the box from each figure.

DISCUSSION

The authors make a good discussion with comparison of results and their interpretation. However, as we have commented in the introduction, the influence of the characteristics of these athletes and this sport may have on the results is missing.

MORE RELEVANT INFORMATION

"We also did not control the athlete’s training outside of the four sessions". Esto es un aspecto muy importante a tener en cuenta, sobre todo al tratarse de una muestra tan pequeña. Los autores deberían plantearse describir (aunque sea de manera resumida) the athlete’s training en el manuscrito ya que podría ser determinante para entender algunas hayazgos encontrados.

A section on practical applications should be included. Also, explain in a clearer and simpler way how this protocol can favor the performance of athletes.

Future lines of research should not go into conclusions. In this section, only detail the most relevant findings of the study. Finally, detail the practical applications.

REFERENCES

Please check the references because in some of them the DOI is not specified and others appear wrongly placed in the manuscript:

L54: Hazell and colleagues (2010)

L104: Brocherie and colleagues (2017)

L 213: (Hamlin et al., 2017)

Author Response

Thank-you very much for your thorough review.

Reviewer 2:

INTRODUCTION

The authors present a complete introduction on the physiological adaptations of training in hypoxic conditions. However, it is necessary to make a detailed explanation of the sport practiced by seven kayakers. Knowing in detail the physiological and competitive requirements of this sport can help understand the adaptations during the study. According to the literature, it seems that not all adaptations in all types of athletes are the same.

Thank you for this recommendation. We have added a few lines in the latter stages of the introduction on the physiological demands of competitive kayak. Remembering canoe and kayak have events ranging between a sorting of < 30 sec (sprint)  to events of 1-2 min duration (1000m). Sp whilst there is a large aerobic component to these events especially 1000m, In both cases especially the sprint there is a heavy anaerobic component. Competitive athletes undertake sprint and other other anaerobic training forms as preparation . Therefore this study is set in that context - comparing the acute responses to  sprint training in hypoxia and normoxia in a group of athletes where anaerobic type training is part of their preparations.

With regards to the comment ‘According to the literature, it seems that not all adaptations in all types of athletes are the same’. We would agree with this comment. Although in the context of this study, it is unclear as to what the reviewer is referring to with this statement

The paper does not clearly explain its advantages concerning the literature: it is not clear the novelty and contributions of the proposed work: does it propose a new method? Or does the novelty only consist of the application?

Thank you for this comment. The introduction section  has been reviewed.

 The novelty of this study is clear and this has been mentioned in paragraph 3 of the introduction and framed by a comprehensive review of literature.

‘The acute response to different durations of anaerobic work intervals performed in hypoxia, controlling for work volume and work-to-rest ratio, has not been previously investigate’

Specifically the comparison  of the acute response (including generation of peak power during each repetition) to different forms of anaerobic training (long and short repetitions) in hypoxia vs normoxia in well trained athletes . This has not yet been done and provides important information about how INTENSITY and DURATION of sprint repetition may influence the (anaerobic glycolytic) response to training in athletes undertaking anaerobic training in hypoxia .this is mentioned on Line 30 and throughout the second paragraph of the introduction

MATERIALS AND METHODS

Eight participants are a very low number of participants. Please add information regarding the sample calculation and the power of your analysis.

Thank-you for your comment. Yes, we do acknowledge that the sample size is low. For this study we had an opportunity to implement a novel training regime in a highly trained population, which made the number of participants involved low in nature due to the special population we were working with. As such, there were no other available athletes within this demographic to participate in order to make our sample size larger. As such, there was no sample size or power calculation completed in advance.  Instead, we included effect size stats.

There is no statement of parental consent concerning the underage kayakers participated in the study?

Thank-you for your comment. All participants were over 18 years of age and did not need parental consent to participate in the study. This was approved by the ethical board at the Canadian Sport Institute Ontario as cited in the manuscript.

What is the study design? Please use a simple diagram or figure to illustrate this paper's whole idea, and the modification it has been made from previous work or traditional framework.In addition, it is necessary to defend the type of design according to the number of subjects and other variables to verify the validity of the study.

Thank-you for your comment. The study design was a randomized blinded crossover design which allowed each athlete to complete each sprint session type (‘lactic’ and ‘alactic ‘) in hypoxia AND normoxia. We believe that this was the best study design for a small sample size - each athlete completed each training session and were blinded to the environmental condition (NOR vs. HYP). Respectively, we believe this doesn’t warrant a figure as the study is quite a simple design. 

What guidelines should athletes follow during protocol sessions and training? Were they only informed both verbally and in writing of the experimental protocol and potential risks? Were measurements made at the same time of day? What about in relation to training? Were there instructions to not train the day of testing prior to assessment? What about hydration status, was this measured prior to assessments?

The participants were informed about the study design and specifics BOTH with an information document and verbally. Verbal explanation was provided as well as questions raised by participants were answered

The written document also contained guidelines on optimal diet prior to exercise (see pony below)

The participants are required to record their diet intake in the 24 hour period prior to the first training session. That diet log was reviewed by the researcher and it was then explained to the participant that they needed to follow the exact same diet intake and pattern including the timing for each subsequent, that is three follow-up sessions. So whilst each participant was given guidelines on principles of high carbohydrate feeding prior to exercise diet intake was not necessarily modified but controlled.

In the same way as diet was recorded. The participants were o’clock for us to not complete any high intensity exercise in the 48 hour period prior to each session they are advised to complete a low intensity session of less than 60 minutes duration in the 48 to 24 hours prior to the first exercise session this was also a requirement for subsequent sessions.

Why did you only analyze peak power output and not mean power output or total distance?

For an exercise protocol of this nature which is anaerobic with the key outcome being generation of the highest power point we felt that peak power is the most important metric for analysis. Indeed the coaching requirement of these seeeion to produce high power output therefore peak power is the most important metric of interest.

L138: L94: SPSS v.24 manufacturer details: IBM Corp., Armonk, NY.

Thank-you very much. This has been added to the manuscript.

"Cohen d are presented to show the magnitude of effect". The authors must specify the thresholds, reference them in "Statistical Analysis" and detail them in all the results.

Thank-you for your comment.  This statement has been added to the Statistical Analysis section, “Effect sizes (η2) were calculated to supplement important findings as the ratio of the mean difference to the pooled SD of the difference. The magnitude of the effect size was classed as trivial (<0.2), small (0.2-0.6), moderate (0.6-1.2), large (1.2-2.0), and very large (>2.0) based on previous published guidelines.”

RESULTS

Please edit the table according to the journal format.

Thank-you, this has been changed.

All the data in text and table must go with a decimal, except the values of "p".

Thank-you for your comment. We are a bit confused with your comment above. Would you please clarify this for us.

The font of the figures must match the font of the text. Also include the vertical axis line and consider removing the box from each figure.

Thank-you. The font on the Table has been changed.

DISCUSSION

The authors make a good discussion with comparison of results and their interpretation. However, as we have commented in the introduction, the influence of the characteristics of these athletes and this sport may have on the results is missing.

Thank-you for your comment.  We have updated the introduction and discussion, and we believe we have addressed this comment (see points above )

MORE RELEVANT INFORMATION

"We also did not control the athlete’s training outside of the four sessions". Esto es un aspecto muy importante a tener en cuenta, sobre todo al tratarse de una muestra tan pequeña. Los autores deberían plantearse describir (aunque sea de manera resumida) the athlete’s training en el manuscrito ya que podría ser determinante para entender algunas hayazgos encontrados.

Thank-you for your comment. We believe we addressed this in the subsequent sentence: “We asked athletes to record modality, intensity, and duration of their training and maintain their training schedules consistent throughout the study. However, adherence to instructions is not always high and it is possible that supplemental training affected the athlete’s ability to perform maximally.” Ultimately, there was no significant difference in training load between the athletes as they were from a homogenous training group.

A section on practical applications should be included. Also, explain in a clearer and simpler way how this protocol can favor the performance of athletes.

Thank-you for your comment. We added a section in the Discussion, Future Directions and Practical Implications, which we believe provides a clear and simpler way to explain the applicability of the results.

Future lines of research should not go into conclusions. In this section, only detail the most relevant findings of the study. Finally, detail the practical applications.

Thank-you for your comment.  The change has been made and the lines about future research has been moved to the Future Directions and Practical Applications section.

REFERENCES

Please check the references because in some of them the DOI is not specified and others appear wrongly placed in the manuscript:

L54: Hazell and colleagues (2010)

L104: Brocherie and colleagues (2017)

L 213: (Hamlin et al., 2017)

Thank-you for your comment. We have updated the references in the document and in the reference list.

Additional references

Michael JS Rooney KE Smith RM (2008) The Metabolic Demands of Kayaking: A Review. Journal of Sports Science & Medicine 7(1):1-7

Thank-you for this reference and our apologies for the oversight. We have added this fantastic reference to the introduction and to the reference list. 

Round 2

Reviewer 2 Report

The authors have made most of the changes. Some of them have not been carried out and adequately justified. However, the following comments need to be addressed:

Comment 3: Regardless of the justification, they can carry out and include he power of your analysis:

https://www.mdpi.com/1660-4601/17/23/8981

Comment 4: I do not understand the answer because in the manuscript you say that the kayakers are between 15 and 24 years old.

Comment 5: You must include a figure with design and protocol characteristics for a better understanding at first sight.

Comment 6: This information must be included in the manuscript. If the variables requested by the reviewer were not considered, they must be added in limitations.

With the previous comments I do not see any objection to the publication of the manuscript.

Author Response

Thank-you for your review!

The authors have made most of the changes. Some of them have not been carried out and adequately justified. However, the following comments need to be addressed:

Comment 3: Regardless of the justification, they can carry out and include the power of your analysis:

https://www.mdpi.com/1660-4601/17/23/8981

Thank-you for your comment and link to the paper.  We are uncertain of how to run a G*3 power calculation on a metric whereby the change with an intervention is currently unknown. At the moment, there are no known articles that are similar enough to compare.  Respectfully, power calculations are related to parametric statistics which assumes the probability (p=…) of an event in a given population.  Since this is a novel intervention with no known similar research using these interventions to compare to, we decided to use the interpretation of effect size as it is a more contemporary method.

Comment 4: I do not understand the answer because in the manuscript you say that the kayakers are between 15 and 24 years old.

Thank-you for your comment. We have changed the language in the Methods to reflect the assent required for participants <18 yrs of age.  This is what the language was changed to “Parents (assent was required) and participants were informed both verbally and in writing of the experimental protocol and potential risks before giving their verbal and written assent and consent, respectively, to participate. The Research Ethics Board at the Canadian Sport Institute Ontario approved the study (REB#18-01).”

Comment 5: You must include a figure with design and protocol characteristics for a better understanding at first sight.

Thank-you. Figure 1 – Study Timeline has been created and uploaded as a PDF. 

Comment 6: This information must be included in the manuscript. If the variables requested by the reviewer were not considered, they must be added in limitations.

With the previous comments I do not see any objection to the publication of the manuscript.

Many thanks for taking the time to review our manuscript and for your invaluable comments.